# Association of healthcare fragmentation with three-year survival among kidney transplant recipients in Colombia

Luis Manuel Barrera-Lozano[1,2], Daniela Sánchez-Santiesteban[1,3], Giancarlo Buitrago [1,3]*

1 Fundación Cardioinfantil – Instituto de Cardiología, Bogotá D.C., Colombia, 2 Sección cirugía de trasplantes, Facultad de Medicina, Universidad de Antioquia, Medellín, Colombia, 3 Instituto de Investigaciones Clínicas, Facultad de Medicina, Universidad Nacional de Colombia, Bogotá, Colombia

* gbuitragog@unal.edu.co

## Abstract

### Objective

Kidney transplantation requires a multidisciplinary approach to achieve optimal outcomes. Healthcare fragmentation, defined by inadequate communication and lack of integration among providers, disrupts the continuum of care, leading to adverse clinical outcomes. Latin American countries face significant challenges in delivering integrated healthcare. This fragmentation is further exacerbated in Colombia by the decentralized structure of the healthcare system, which disperses responsibilities across multiple actors, including public and private providers, insurers, and regional authorities. This study aimed to assess healthcare fragmentation in kidney transplant patients during their first post-transplant year and evaluate its association with three-year survival among patients enrolled in Colombia's contributory healthcare scheme.

### Methods

A retrospective cohort study was conducted using administrative data from Colombia's contributory healthcare scheme. The cohort included kidney transplant recipients (2012–2016) who survived the first post-transplant year. Healthcare fragmentation was measured by the number of unique providers involved in the first year. Patients were categorised into high- and low-fragmentation groups based on the 75th percentile of provider distribution. The primary outcome was three-year survival, analysed using multivariate Cox regression to estimate hazard ratios (HRs), adjusted for age, sex, Charlson Comorbidity Index (CCI), insurer, region, and transplant year.

### Results

The cohort comprised 2,028 kidney transplant patients, with a mean age of 47.7 years (SD: 13.4), 38.7% female, and 68.7% presenting a CCI ≤ 3. Healthcare

**Data availability statement:** The following information sources: Single Registry of Enrollees, Mortality Registry Module from the Unified Affiliation Registry (RUAF) and Calculation Study of the Capitation Unit Database (Base del Estudio de Suficiencia de la Unidad Por Capitación, or UPC) are administered by the Colombian Ministry of Health and Social Protection. These databases are freely available upon request to the Technology of the Information and Communication Office of the Colombian Ministry of Health and Social Protection through the e-mail: correo@minsalud.gov.co.

**Funding:** This study was partially funded by the NIHR GHPSR researcher-led grant NIHR150067, which used UK aid from the UK Government to support global health research. Authors were also supported by Fundación Cardioinfantil – Instituto de Cardiología. The funders did not contribute to study design, fieldwork, data analysis, decision to publish or preparation of the manuscript.

**Competing interests:** The authors have declared that no competing interests exist.

fragmentation ranged from 1 to 34 providers, with a mean of 8.94 (SD: 6.77). High fragmentation (≥11 providers) was observed in 30.2% of patients. Three-year mortality was significantly higher in the high-fragmentation group (18%) compared to the low-fragmentation group (12%) (p = 0.04). High fragmentation was associated with a 49% increased mortality risk (adjusted HR: 1.49; 95% CI: 1.12–1.97; p = 0.01).

## Conclusion

These findings underscore the importance of integrated care models and improved coordination among providers to enhance patient outcomes, particularly in resource-limited settings.

## Introduction

Kidney transplantation is recognised as the most effective treatment for patients with end-stage renal disease, significantly improving survival rates and quality of life compared to long-term dialysis [1,2]. Despite its clinical benefits, the success of renal transplantation depends heavily on comprehensive, multidisciplinary care that addresses various challenges, including immunosuppressive therapy, management of comorbid conditions, and the prevention and early detection of complications [3–5]. Coordinated care is critical in ensuring optimal outcomes, as patients require regular follow-ups, monitoring by specialised teams, and laboratory by healthcare providers [5]. However, fragmentation within healthcare systems, defined by inadequate communication and lack of integration among providers, disrupts this continuum of care, leading to adverse clinical outcomes [6–9]. In kidney transplantation, for example, fragmentation can lead to delays in diagnosis and treatment, duplication of medical services, increased costs, and worse health outcomes, ultimately compromising the potential benefits of transplantation [10,11].

Due to their fragmented health systems, Latin American countries face significant challenges in delivering integrated healthcare. Colombia, despite achieving universal health coverage through contributory (formal employees and their families) and subsidised (informal workers and their families) schemes covering over 97% of the population [12], faces persistent challenges related to healthcare fragmentation. In Colombia, fragmentation manifests as a lack of coordination both across providers at the same level of care (horizontal fragmentation) and between different levels of care and administrative entities such as insurers and public institutions (vertical fragmentation) [13]. This fragmentation is further exacerbated in Colombia by the decentralized structure of the healthcare system, which disperses responsibility across multiple actors, including public and private providers, insurers, and regional authorities. For complex medical procedures such as kidney transplantation, once the surgery is completed, the insurer determines which providers will deliver follow-up care, including medical appointments, laboratory tests and follow-up services. Some insurers maintain agreements with specialized centers, allowing for care within a single institution during the first 6–12 months post-transplant. Others, however, lack

such arrangements, resulting in patients receiving care from multiple providers. In these cases, the consequences of fragmentation are particularly severe as they encounter obstacles such as inconsistent follow-up, and insufficient communication between transplant centres and other providers. These challenges not only hinder recovery but also increase the likelihood of complications, hospital readmissions, and preventable mortality.

Healthcare fragmentation has been studied in chronic conditions such as cancer and cardiovascular diseases [14–17]; however, kidney transplantation, which requires a similarly high level of care integration, remains understudied. To our knowledge, no studies have specifically evaluated the association between healthcare fragmentation and survival outcomes among kidney transplant recipients in either high-income countries (HICs) or low- and middle-income countries (LMICs). Some studies have explored fragmentation in other transplant populations, such as liver transplant recipients, and have reported adverse outcomes, including increased mortality and risk of rehospitalization [18,19]. This gap in the literature highlights the need for focused research on how fragmented care affects kidney transplant outcomes, particularly in settings like Colombia, where systemic fragmentation is a well-documented barrier to effective healthcare delivery.

To our knowledge, no studies have specifically evaluated the association between healthcare fragmentation and survival outcomes in renal transplant recipients in high income countries (HIC), nor low- and middle-income countries (LMICs). This gap in the literature underscores the need for targeted research to understand better how fragmented care influences transplant success, particularly in the Colombian context, where fragmentation challenges are pronounced.

This study aims to address this gap by assessing the association of care fragmentation experienced by number of providers that attended renal transplant patients during the first year following transplantation and the three-year overall survival. The findings can potentially inform health policies and interventions to reduce care fragmentation, improve coordination among healthcare providers, and optimise resource use in this population. Ultimately, the study would highlight the importance of integrated care models in enhancing renal transplant patients' survival, particularly in resource-limited and fragmented healthcare systems like Colombia.

## Methods

### Ethics

This study was granted institutional review board (IRB) ethical approval in 10/03/2022 by the Faculty of Medicine Ethics Committee at the Universidad Nacional de Colombia (Approval Number: 004–029). Written consent was waived by the IRB as data sources were administrative databases fully anonymised.

The databases were provided to the Clinical Research Institute at the Faculty of Medicine of the Universidad Nacional de Colombia by the Office of Information Technologies of the Ministry of Health (MoH) for research purposes, S1 Text. Prior to their transfer, the Ministry anonymised the data by generating a unique identifier that allows for linkage across databases and enables longitudinal follow-up of individuals. Following ethics approval, initial access to the databases was granted from 09/07/2022–02/09/2022 with the purpose of identifying patients who underwent renal transplantation, cleaning the database by selecting a sample of patients meeting the eligibility criteria, and creating a refined dataset for subsequent analysis. This refined dataset forms the basis of the analyses presented in this paper. This project was presented at the XI Congreso Colombiano de Trasplantes de Órganos, held in Barranquilla, Colombia in August 2022 and was awarded third place in the category of Best Original Work.

### Study design and population

This retrospective cohort study was conducted based on administrative data of adult patients who underwent kidney transplantation between 2012 and 2016 and were enrolled in the contributory health insurance system. The cohort was constructed using health claims data from the Base for the Study of Capitation Unit Sufficiency (UPC) database to identify all adult patients with a kidney transplantation record [20]. All subjects included in the study had to survive the first-year

post-transplant to allow for the measurement of fragmentation during this period. After this initial year, each subject was followed for up to three years or until death, whichever occurred first. We used the Mortality Registry Module from the Unified Affiliation Registry (RUAF) to determine the mortality date [21].

## Data source

The UPC database is highly standardised and is the primary source of information used by the MoH for the annual estimation of the UPC in the healthcare system. The database includes detailed information on each utilised service, such as the Unified Health Procedure Code (CUPS), an ICD-10 code associated with the service, the service date, the cost paid by each insurer to each provider, the patient's sex, the insurer to which the patient is affiliated, the city where the service was provided, the provider's registration code, and an anonymised individual identifier for each affiliate. Additionally, the RUAF Mortality Registry Module contains information from death certificates for all Colombians, including the date of death and diagnosis. International assessments have confirmed the reliability of RUAF data, with 91% of deaths registered through death certificates as of 2016 [22].

## Exposure, outcome, and control variables

As mentioned, all included patients survived the first-year post-transplantation as an inclusion criterion. To measure healthcare fragmentation, we counted the number of unique healthcare providers involved in a patient's care during the first year following kidney transplantation. This approach, used in prior studies, serves as a proxy for fragmented care by reflecting the dispersion of services across different providers [14–16,23,24]. Each provider code corresponds to a distinct institution or healthcare facility as recorded in the administrative claims database. While this metric does not capture coordination quality directly, higher counts are assumed to indicate lower care integration. We included all types of services, beyond transplant-specific care, to capture the full scope of medical attention required for these patients. This approach reflects the complexity of managing comorbidities and the need for integrated care. After the first year of survival, all patients were followed for an additional three years. Using information from the RUAF database, the three-year survival time was estimated, corresponding to the study's primary outcome variable. Control variables included age, sex, insurer, geographic region, and year of transplantation and comorbidities measured using the Charlson Comorbidity Index (CCI) [25]. The CCI was validated for use in Colombia by Oliveros et al. [25] derived from ICD-10 codes. The CCI considered the presence of the following conditions: acute myocardial infarction, congestive heart failure, peripheral vascular disease, stroke, dementia, chronic pulmonary disease, connective tissue disease, peptic ulcer disease, liver disease, diabetes, complications of diabetes, cancer, metastatic cancer, paraplegia, renal disease, severe liver disease, and HIV.

## Analysis

Baseline characteristics of the cohort were described using measures of central tendency and dispersion for continuous variables and relative and absolute frequencies for categorical variables. The quartiles of the primary exposure variable (i.e., the number of different providers in the first-year post-transplantation) were identified.

The cohort was divided into high fragmentation (those exposed to a number of providers greater than or equal to the 75th percentile of the distribution) and low fragmentation (those exposed to a number of providers below the 75th percentile), this threshold was chosen to identify individuals exposed to a substantially higher-than-average degree of fragmentation. Similar percentile-based approaches have been used in health services research to define extreme exposure groups [14–16]. An analysis of the crude three-year mortality proportion and unadjusted survival was performed. Using a multivariate Cox regression model, the hazard ratio (HR) of being exposed to highly fragmented healthcare compared to low-fragmentation healthcare was estimated, controlling for the previously mentioned covariables. The proportionality assumption was evaluated with graphical and statistical tests using Schoenfeld residuals. All estimators were presented with 95% confidence intervals. Analyses were conducted using Stata 17 MP (licensed to the Universidad Nacional de

Colombia). This article followed the STROBE (Strengthening the Reporting of Observational Studies in Epidemiology) [26] guidelines to ensure transparency and thoroughness in reporting the findings, S1 Checklist.

## Results

Between 2012 and 2016, a total of 3,999 patients underwent kidney transplantation. After excluding 47 patients under 18 years old and 1,924 patients who died within the first year, 2,028 patients were eligible for the study. No records were lost in the follow-up as each of the 2,028 participants was followed for three years or until death, whichever occurred first. The flowchart in Fig 1 provides additional details on the selection process.

Table 1 presents detailed sociodemographic, clinical and exposure characteristics of the cohort. Of the total cohort, 785 patients (38.7%) were women, and the mean age was 47.7 years (Standard Deviation [SD] 13.4 years). A total of 1,394 patients (68.7%) had a CCI of 3 or less. Bogotá was the region with the highest number of transplants, with a total of 923 (45.5%), and a decline in the number of transplants per year was observed (see Table 1).

The degree of healthcare fragmentation, measured as the number of distinct providers involved in a patient's care during the first year post-transplant, ranged from 1 to 34, with a mean of 8.94 (SD: 6.77). The 25th, 50th, and 75th percentiles were 6, 8, and 11 providers, respectively. Among the regions analysed, the Central region exhibited the highest mean fragmentation, with an average of 10.78 providers (SD: 4.65), while the Pacific region showed the lowest, with an average of 7.96 providers (SD: 3.73). (Table 2). At the departmental level, the highest fragmentation was observed in Sucre (14.00; SD: 5.10), La Guajira (12.13; SD: 5.67), and Caquetá (13.60; SD: 5.19). Conversely, the lowest fragmentation was found

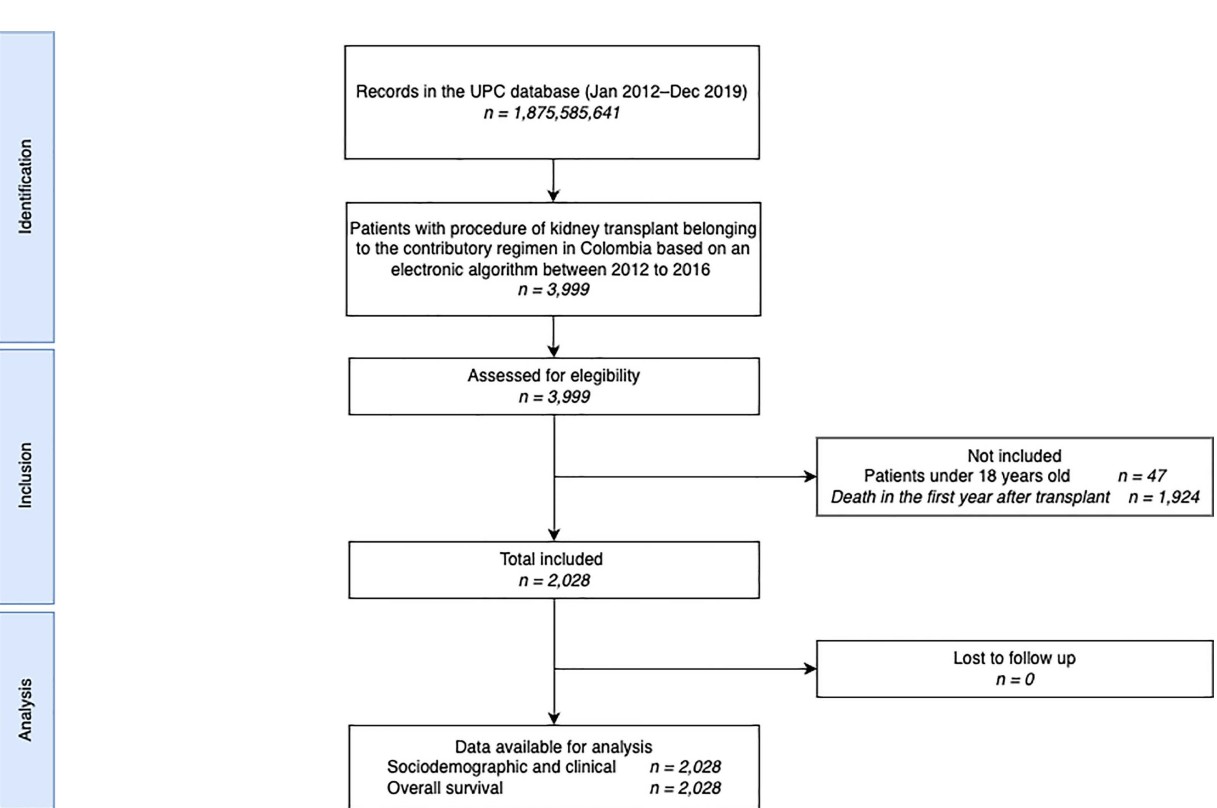

**Fig 1. STROBE flow chart.** UPC: Base for the Study of Capitation Unit Sufficiency Database.

**Table 1. Description of cohort baseline characteristics.**

| Variable | Total | Level of healthcare fragmentation | |
|---|---|---|---|
| | | Low | High |
| | N = 2,028 | N = 1,415 | N = 613 |
| Female; N (%) | 785 (38.7%) | 543 (38.4%) | 242 (39.5%) |
| Age (years); X (SD) | 47.70 (13.45) | 47.26 (13.43) | 48.72 (13.46) |
| Categorised age; N (%) | | | |
| 18-30 | 237 (11.7%) | 167 (11.8%) | 70 (11.4%) |
| 30-49 | 829 (40.9%) | 605 (42.8%) | 224 (36.5%) |
| 50-59 | 560 (27.6%) | 378 (26.7%) | 182 (29.7%) |
| 60-69 | 341 (16.8%) | 221 (15.6%) | 120 (19.6%) |
| 70 or older | 61 (3.0%) | 44 (3.1%) | 17 (2.8%) |
| CCI; Median (IQR) | 3.00 (2.00) | 2.00 (2.00) | 3.00 (3.00) |
| Categorised CCI; N (%) | | | |
| 3 or less | 1,395 (68.8%) | 1,034 (73.1%) | 361 (58.9%) |
| 4 a 5 | 411 (20.3%) | 240 (17.0%) | 171 (27.9%) |
| 6 a 7 | 151 (7.4%) | 95 (6.7%) | 56 (9.1%) |
| 8 or more | 71 (3.5%) | 46 (3.3%) | 25 (4.1%) |
| Geographic Region[a]; N (%) | | | |
| Atlantica | 163 (8.0%) | 84 (5.9%) | 79 (12.9%) |
| Metropolitana | 923 (45.5%) | 697 (49.3%) | 226 (36.9%) |
| Central | 377 (18.6%) | 201 (14.2%) | 176 (28.7%) |
| Oriental | 183 (9.0%) | 119 (8.4%) | 64 (10.4%) |
| Pacífica | 373 (18.4%) | 307 (21.7%) | 66 (10.8%) |
| Orinoquía and Amazonía | 9 (0.4%) | 7 (0.5%) | 2 (0.3%) |
| Year of kidney transplant; N (%) | | | |
| 2012 | 571 (28.2%) | 348 (24.6%) | 223 (36.4%) |
| 2013 | 430 (21.2%) | 299 (21.1%) | 131 (21.4%) |
| 2014 | 493 (24.3%) | 414 (29.3%) | 79 (12.9%) |
| 2015 | 291 (14.3%) | 208 (14.7%) | 83 (13.5%) |
| 2016 | 243 (12.0%) | 146 (10.3%) | 97 (15.8%) |
| Healthcare fragmentation on first year post- kidney transplant | | | |
| Number of healthcare providers; X (SD) | 8.94 (4.47) | 6.63 (2.25) | 14.27 (3.71) |
| Categorised Healthcare fragmentation; N (%) | | | |
| Low fragmentation (10 or less) | 1,415 (69.8%) | 1,415 (100.0%) | 0 (0.0%) |
| High fragmentation (11 or more) | 613 (30.2%) | 0 (0.0%) | 613 (100.0%) |

Abbreviations: CI: Confidence Interval. N: Number; X: Mean; SD: Standard Deviation; IQR: Interquartile Range; CCI: Comorbidities Charlson Index.

[a]Atlántica: Atlántico, Bolívar, Cesar, Córdoba, Magdalena, La Guajira, Sucre, and San Andrés; Metropolitan: Bogotá D.C. and Cundinamarca; Central: Antioquia, Caldas, Huila, Quindío, Risaralda, and Tolima; Oriental: Arauca, Boyacá, Caquetá, Casanare, Meta, Norte de Santander, Santander, and Vichada; Pacífica: Cauca, Chocó, Nariño, Putumayo, and Valle del Cauca; Orinoquía and Amazonía: Amazonas, Guainía, Guaviare, and Vaupés.

in Nariño (6.33; SD: 2.65) and Arauca (6.50; SD: 2.12). Fig 2 illustrates the geographic distribution of healthcare fragmentation across Colombia, highlighting significant regional variability.

Patients were categorised into two groups: high fragmentation, comprising those who consulted 11 or more providers (≥p75), and low fragmentation, comprising those who consulted ten or fewer providers (<p75). 1,415 patients (69.7%) experienced low fragmentation, whereas 613 patients (30.2%) were exposed to high fragmentation. Table 3 shows the

**Table 2. Healthcare Fragmentation by Geographic Regions in Colombia according to the number of different providers involved during the first year of post-transplant healthcare attention.**

| Geographic Region | Mean | 95% CI |
|---|---|---|
| Atlántica[a] | 10.65 | 7.92; 13.38 |
| Bogotá DC[b] | 8.25 | 5.51; 11.00 |
| Central[c] | 10.79 | 7.90; 13.67 |
| Oriental[d] | 9.07 | 6.42; 11.72 |
| Pacífica[e] | 7.97 | 5.66; 10.28 |
| Orinoquía and Amazonía[f] | 9.22 | 6.73; 11.72 |

Abbreviations: CI: Confidence Interval.

[a]Atlántica: Atlántico, Bolívar, Cesar, Córdoba, Magdalena, La Guajira, Sucre, and San Andrés; [b]Metropolitan: Bogotá D.C. and Cundinamarca; [c]Central: Antioquia, Caldas, Huila, Quindío, Risaralda, and Tolima; [d]Oriental: Arauca, Boyacá, Caquetá, Casanare, Meta, Norte de Santander, Santander, and Vichada; [e]Pacífica: Cauca, Chocó, Nariño, Putumayo, and Valle del Cauca; [f]Orinoquía and Amazonía: Amazonas, Guainía, Guaviare, and Vaupés.

crude three-year mortality proportions for each baseline characteristic evaluated. Patients exposed to low fragmentation had a mortality proportion of 0.12, while those exposed to high fragmentation had a mortality proportion of 0.18 (p = 0.04), consistent with the estimates from the unadjusted Cox model (HR: 1.49 (95% IC: 1.17–1.90), p = 0.001). Other characteristics where differences were observed included age category and CCI. No differences were found by sex or geographic region.

Table 4 presents the multivariate Cox model, showing that exposure to a highly fragmented healthcare increases the risk of death at 3 years by 49% (HR: 1.49; 95% CI 1.12–1.97; p = 0.01) after adjusting for sex, age, CCI, geographic region, year of transplantation, and insurer (Fig 3). Other factors associated with lower three-year survival included being over 30 years old, having a CCI greater than 4, and being female. No violation of the proportional-hazards assumption was detected. The global Schoenfeld test yielded $X^2 = 16.89$ (df = 23, p = 0.815) and no covariate showed an individual p-value < 0.05.

## Discussion

This study evaluated the association between healthcare fragmentation during the first year after kidney transplantation and survival outcomes. Our findings demonstrate that high fragmentation, defined as exposure to 11 or more unique healthcare providers within the first-year post-transplant, was associated with a 49% increased mortality risk within three years (HR: 1.49; 95% CI 1.12–1.97; p = 0.01) for each additional provider participating in healthcare during the first-year post-transplant. This suggests that a greater number of providers after undergoing kidney transplant, in the absence of effective care coordination, may compromise continuity of care and adversely impact on patients' survival.

Healthcare fragmentation in this study was measured by the number of different healthcare institutions involved in a patient's care during the first-year post-transplant. To our knowledge, this is the first study to examine the association between healthcare fragmentation and three-year survival among kidney transplant recipients in both low- and high-income country settings. Previous studies in transplant populations have primarily focused on the impact of appointment adherence on graft function and patient survival. However, to date, no research has evaluated care from a healthcare system perspective, specifically addressing fragmentation as done in our study. While healthcare fragmentation has been explored in other chronic conditions, such as liver transplantation and cancer, our findings are consistent with prior literature showing that higher fragmentation is associated with lower survival rates and increased healthcare costs [14,15,23,24]. Mechanisms driving these associations likely include delays in treatment, duplication of medical services, and poor coordination among healthcare providers, all of which undermine the potential benefits of transplantation [17].

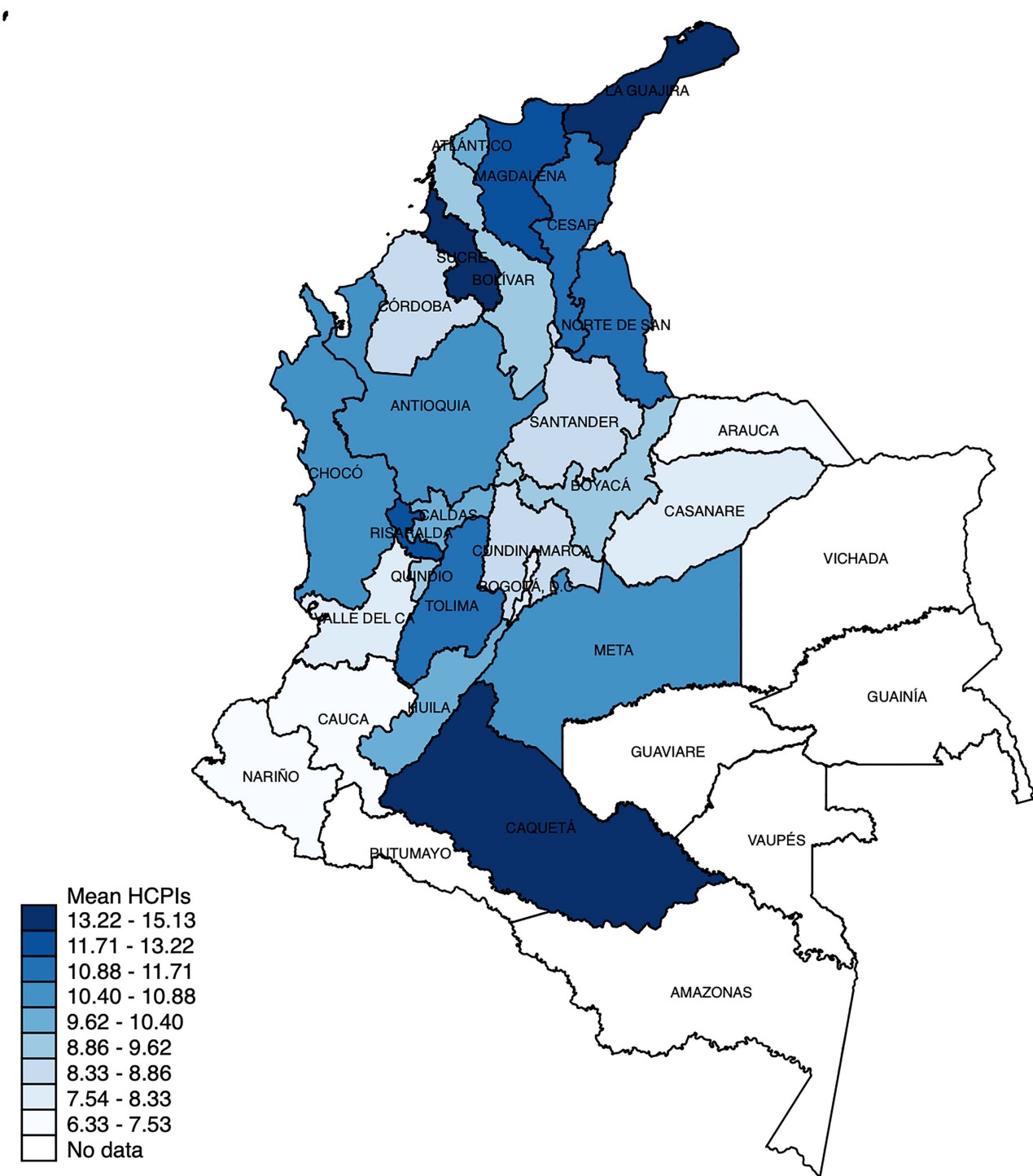

**Fig 2. Geographical distribution of healthcare fragmentation.** Geographic distribution of exposure to healthcare fragmentation measured as number of different HCPIs in the first year of care after kidney transplantation in adults belonging to the contributory regimen in Colombia. Abbreviations: HCPIs: Healthcare Provider Institutions.

**Table 3. Three-year Mortality Proportions by Baseline Characteristics.**

| Characteristic | Mortality Proportion | 95% CI |
|---|---|---|
| Fragmentation in the first year post-transplant | | |
| Low fragmentation (10 or fewer providers) | 0.12 | 0.10 - 0.14 |
| High fragmentation (11 or more providers) | 0.18 | 0.14 - 0.21 |
| Sex | | |
| Female | 0.14 | 0.12 - 0.16 |
| Male | 0.14 | 0.11 - 0.16 |
| Age category | | |
| 18-30 | 0.04 | 0.01 - 0.06 |
| 30-49 | 0.09 | 0.07 - 0.11 |
| 50-59 | 0.15 | 0.12 - 0.18 |
| 60-69 | 0.27 | 0.21 - 0.32 |
| 70 or older | 0.39 | 0.24 - 0.55 |
| Charlson Comorbidity Index | | |
| 3 or fewer | 0.10 | 0.08 - 0.11 |
| 4–5 | 0.23 | 0.19 - 0.28 |
| 6–7 | 0.25 | 0.17 - 0.33 |
| 8 or more | 0.14 | 0.05 - 0.23 |
| Geographic region | | |
| Atlantic | 0.09 | 0.05 - 0.14 |
| Bogotá DC | 0.16 | 0.13 - 0.19 |
| Central | 0.11 | 0.08 - 0.15 |
| Eastern | 0.16 | 0.10 - 0.22 |
| Pacific | 0.13 | 0.09 - 0.16 |
| Other departments | 0.00 | NA |
| Year of transplant | | |
| 2012 | 0.12 | 0.09 - 0.15 |
| 2013 | 0.11 | 0.08 - 0.14 |
| 2014 | 0.17 | 0.14 - 0.21 |
| 2015 | 0.15 | 0.10 - 0.19 |
| 2016 | 0.15 | 0.10 - 0.20 |

Abbreviations: 95% CI, 95% confidence interval. *: statistically significant (p<0.05).

Our results also revealed significant geographical variability in fragmentation across Colombia. Patients in the Central and Atlantic regions experienced the highest levels of fragmentation. In contrast, patients in the Pacific region exhibited lower fragmentation levels, likely due to limited access to specialised care rather than effective healthcare integration. Additionally, in areas with low population density and fewer healthcare facilities, patients often need to travel long distances to access specialised care, potentially leading to delays in treatment. These findings underscore the need for fragmentation metrics that account for travel distance and geographic barriers, particularly in countries with pronounced disparities in healthcare availability and accessibility.

These findings have direct implications for health policy in countries with fragmented healthcare delivery systems, such as Colombia. Our results underscore the urgent need to implement coordinated care models and standardized referral pathways for kidney transplant recipients. In Colombia, since 2015, national efforts have been underway to standardize the management of priority health conditions through the implementation of Comprehensive Healthcare Service Pathways

**Table 4. Adjusted HR of overall survival of adults with kidney transplants by the level of healthcare fragmentation, defined by the number of different providers involved during the first year of post-transplant healthcare attention.**

| Characteristic | Hazard Ratio | 95% CI | P-Value |
|---|---|---|---|
| Fragmentation in the first year post-transplant | | | |
| Low fragmentation (10 or fewer providers) | Ref | | |
| High fragmentation (11 or more providers) | 1.49 | 1.12 - 1.97 | 0.01* |
| Sex | | | |
| Male | Ref | | |
| Female | 1.30 | 1.01 - 1.67 | 0.04* |
| Age category | | | |
| 18-30 | Ref | | |
| 30-49 | 2.30 | 1.15 - 4.60 | 0.02* |
| 50-59 | 3.88 | 1.93 - 7.80 | <0.01* |
| 60-69 | 6.84 | 3.40 - 13.77 | <0.01* |
| 70 or older | 12.01 | 5.51 - 26.20 | <0.01* |
| Charlson Comorbidity Index | | | |
| 3 or fewer | Ref | | |
| 4–5 | 1.93 | 1.47 - 2.54 | <0.01* |
| 6–7 | 1.83 | 1.24 - 2.69 | <0.01* |
| 8 or more | 1.09 | 0.57 - 2.10 | 0.79 |
| Geographic region | | | |
| Atlantic | Ref | | |
| Bogotá DC | 1.26 | 0.69 - 2.29 | 0.45 |
| Central | 0.93 | 0.51 - 1.71 | 0.82 |
| Eastern | 1.63 | 0.86 - 3.06 | 0.13 |
| Pacific | 1.12 | 0.61 - 2.08 | 0.71 |
| Other departments | 0.00 | 0.00 -. | 1.00 |
| Year of transplant | | | |
| 2012 | Ref | | |
| 2013 | 0.95 | 0.65 - 1.39 | 0.78 |
| 2014 | 1.19 | 0.83 - 1.71 | 0.35 |
| 2015 | 1.18 | 0.78 - 1.78 | 0.43 |
| 2016 | 1.47 | 0.92 - 2.35 | 0.11 |

The HR was adjusted for Sex, Age, CCI, geographic region, year of transplant and insurer. Abbreviations: 95% CI, 95% confidence interval; Ref: Reference. *: statistically significant (p < 0.05).

(Rutas Integrales de Atención en Salud, RIAS). These pathways aim to improve care coordination and ensure continuity across different levels of care. Our study contributes novel evidence from a middle-income country and highlights the need to adapt and extend integration strategies such as RIAS to complex, high-risk populations like kidney transplant recipients.

This study has several strengths. The reliability of the data is a crucial advantage as it provides robust evidence on the relationship between healthcare fragmentation and transplant outcomes. The UPC database has been widely used in previous studies [27–34], and the RUAF death registry offers extensive coverage, documenting over 90% of deaths nationwide, further supporting their validity as valuable research tools. Additionally, the specificity of kidney transplant procedures, which are unlikely to be misclassified due to their cost and complexity, enhances the reliability of the data.

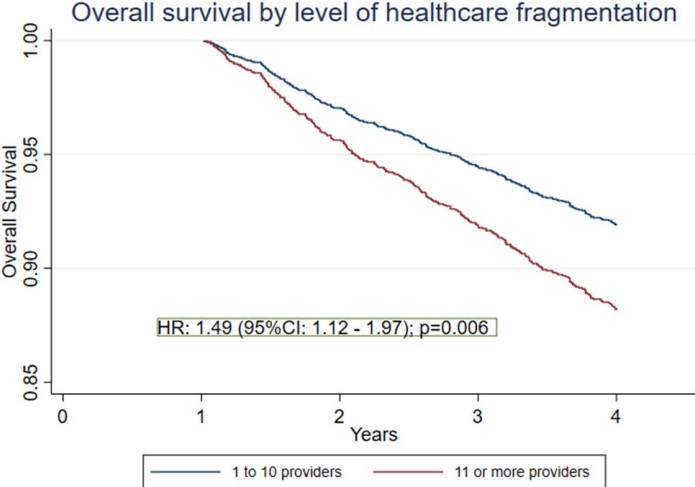

**Fig 3. Three-year overall survival of adults with kidney transplants by the level of healthcare fragmentation, defined by the number of different providers involved during the first year of post-transplant healthcare attention.**

However, some limitations must be addressed. Administrative databases lack detailed clinical information, which may leave behind some crucial covariables. To mitigate this limitation, we incorporated comorbidity data using the CCI, a validated scale for its use with administrative datasets in Colombia. The selection and interpretation of this index were further supported by clinical experts in transplantation. Furthermore, there needs to be a standardised measure of fragmentation that limits comparisons with studies from other parts of the world. Future research should investigate more refined metrics of healthcare fragmentation to better elucidate its relationship with clinical outcomes, as well as the association between fragmentation and survival, stratified by the complexity level of the services used to assess fragmentation. Qualitative studies could also provide valuable insights into the experiences of patients and providers navigating fragmented healthcare systems, helping to inform policy and improve care integration.

## Conclusion

This study demonstrates that higher care fragmentation, measured by the number of unique healthcare providers involved during the first-year post-transplantation, is associated with reduced three-year overall survival in renal transplant patients in Colombia. Patients exposed to high fragmentation faced a 49% increased risk of death, highlighting the critical need to improve care coordination in this population. Our findings emphasise the importance of integrated healthcare models and enhance care coordination to reduce fragmentation, mortality risks, and optimise resource use in kidney transplant patients. Policymakers should prioritise strategies to streamline healthcare delivery, particularly in resource-limited settings and fragmented systems like Colombia. Further research is needed to explore effective strategies for integrating care and addressing regional disparities in healthcare access, which could inform policies to improve outcomes for patients with chronic conditions, including those requiring renal transplantation.

## Supporting information

**S1 Text. Authorization Letter from the Ministry of Health of Colombia for the Use of Administrative Databases in Clinical Research.**
(PDF)

**S1 Checklist. STROBE Statement.**
(DOCX)

## Acknowledgments

We thank Colombia's Ministry of Health and Social Protection for providing the administrative databases that made this study possible. We also thank the Clinical Research Student Group at the Faculty of Medicine of the Universidad Nacional de Colombia for their valuable contribution and La Fundación Cardioinfantil – LaCardio transplant and investigation groups for their valuable support. This study's preliminary findings were presented and discussed within these groups, enriching the interpretation and scope of our results.

## Author contributions

**Conceptualization:** Luis Manuel Barrera-Lozano, Giancarlo Buitrago.

**Formal analysis:** Daniela Sánchez-Santiesteban, Giancarlo Buitrago.

**Investigation:** Luis Manuel Barrera-Lozano, Giancarlo Buitrago.

**Software:** Daniela Sánchez-Santiesteban.

**Writing – original draft:** Daniela Sánchez-Santiesteban.

**Writing – review & editing:** Luis Manuel Barrera-Lozano, Daniela Sánchez-Santiesteban, Giancarlo Buitrago.

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
