## [Decision Letter · Decision Letter 0]

22 Apr 2025

PONE-D-24-57142Association of healthcare fragmentation and overall survival in patients with kidney transplant in ColombiaPLOS ONE

Dear Dr. Buitrago,

Thank you for submitting your manuscript to PLOS ONE. After careful consideration, we feel that it has merit but does not fully meet PLOS ONE’s publication criteria as it currently stands. Therefore, we invite you to submit a revised version of the manuscript that addresses the points raised during the review process.

While the introduction refers to healthcare fragmentation in general terms, it lacks sufficient contextualization regarding how fragmentation specifically affects kidney transplant patients in Colombia. I recommend expanding this section to clarify why Colombia is an appropriate setting for studying healthcare fragmentation and how particular features of its healthcare system contribute to this issue. Strengthening the introduction in this way would enhance the overall framing of the study.

Clearly define what is meant by “fragmentation” within the context of this research. It is essential to specify how fragmentation is conceptualized and measured, and to elaborate on the mechanisms through which it impacts the Colombian healthcare system, particularly for kidney transplant recipients.

The discussion section should be strengthened. I suggest analyzing how the study’s findings compare or contrast with those reported in other countries. This comparative approach would better highlight the study's contribution to the global literature on healthcare fragmentation and transplant outcomes.

Improve the conclusion to make it more assertive and impactful. It should not only summarize the key findings, but also clearly articulate the implications for policy, practice, and future research.

Please submit your revised manuscript by  Jun 06 2025 11:59PM. If you will need more time than this to complete your revisions, please reply to this message or contact the journal office at plosone@plos.org . Please include the following items when submitting your revised manuscript:

We look forward to receiving your revised manuscript.

Kind regards,

Oriana Rivera-Lozada de Bonilla

Academic Editor

PLOS ONE

As you are reporting a retrospective study of medical records or archived samples, please ensure that you have discussed whether all data were fully anonymized before you accessed them and/or whether the IRB or ethics committee waived the requirement for informed consent. If patients provided informed written consent to have data from their medical records used in research, please include this information.

“GB and DSS were partially funded by the NIHR GHPSR researcher-led grant NIHR150067, which used UK aid from the UK Government to support global health research.”

5. For studies involving third-party data, we encourage authors to share any data specific to their analyses that they can legally distribute. PLOS recognizes, however, that authors may be using third-party data they do not have the rights to share. When third-party data cannot be publicly shared, authors must provide all information necessary for interested researchers to apply to gain access to the data. (https://journals.plos.org/plosone/s/data-availability#loc-acceptable-data-access-restrictions)

6. We note that Figure 2 in your submission contain map/satellite images which may be copyrighted. All PLOS content is published under the Creative Commons Attribution License (CC BY 4.0), which means that the manuscript, images, and Supporting Information files will be freely available online, and any third party is permitted to access, download, copy, distribute, and use these materials in any way, even commercially, with proper attribution. For these reasons, we cannot publish previously copyrighted maps or satellite images created using proprietary data, such as Google software (Google Maps, Street View, and Earth). For more information, see our copyright guidelines: http://journals.plos.org/plosone/s/licenses-and-copyright.

Reviewers' comments:

Reviewer's Responses to Questions

**Comments to the Author**

1. Is the manuscript technically sound, and do the data support the conclusions?

Reviewer #1: Yes

Reviewer #2: Partly

Reviewer #3: Yes

Reviewer #4: Yes

2. Has the statistical analysis been performed appropriately and rigorously? 

Reviewer #1: Yes

Reviewer #2: Yes

Reviewer #3: Yes

Reviewer #4: Yes

3. Have the authors made all data underlying the findings in their manuscript fully available?

Reviewer #1: Yes

Reviewer #2: Yes

Reviewer #3: Yes

Reviewer #4: No

4. Is the manuscript presented in an intelligible fashion and written in standard English?

Reviewer #1: Yes

Reviewer #2: Yes

Reviewer #3: Yes

Reviewer #4: Yes

5. Review Comments to the Author

Reviewer #1: Association of healthcare fragmentation and overall survival in patients with kidney transplant in Colombia

Title

• Make the title more engaging and reflective of the results. For example:

"Impact of Healthcare Fragmentation on Three-Year Survival of Kidney Transplant Recipients in Colombia."

Abstract

• The abstract does not mention why healthcare fragmentation is a significant issue in Colombia or LMICs. Add a sentence explaining why fragmentation is relevant in Colombia or LMICs.

• The conclusion is clear but repetitive with the introduction. Streamline the conclusion to avoid repetition. Focus on the implications and recommendations.

Introduction

• While the introduction mentions healthcare fragmentation in general, it does not provide enough context about how fragmentation specifically impacts kidney transplant patients in Colombia. Expand on why Colombia is a relevant setting for studying fragmentation and how its healthcare system's structure creates challenges.

• The phrase "fragmentation manifests both horizontally... and vertically" could be explained more clearly for a general audience.

Methods

• The definition of healthcare fragmentation ("measured by the number of different healthcare providers") is too simplistic and lacks methodological depth. Provide more detail on how fragmentation was quantified.

• The rationale for selecting the 75th percentile as the cut-off for high fragmentation is not explained. Justify the use of the 75th percentile for defining high fragmentation.

Results

• Descriptions of geographical fragmentation are overly detailed, making it difficult to identify key findings. Focus on the most important findings and present them concisely. Focus on the most important findings and present them concisely. For example: "Patients with high fragmentation (≥11 providers) had a significantly higher mortality rate (18%) compared to those with low fragmentation (12%) (p=0.04)."

• Some results, such as "high fragmentation was associated with a 49% increased mortality risk," are presented without interpretation or context. Briefly explain the implications of key results, e.g.: "This suggests that greater provider involvement, without effective care coordination, negatively impacts patient outcomes."

• Some terms (e.g., "fragmentación") appear in Spanish (Table 1), which is inconsistent with the rest of the English manuscript. Standardize the language to ensure all terms are in English.

Discussion

• Some statements, such as "fragmentation leads to poor outcomes," are too broad and not linked directly to the study findings. Link findings more explicitly to practical implications.

• The discussion does not sufficiently address how the findings can be used to inform healthcare policies or improve care coordination in Colombia. Discuss how the findings align or differ from those in other countries to emphasize the study's contribution to global literature.

• The study could benefit from comparing findings with fragmentation studies in high-income countries to provide a broader context. Avoid overly general statements by linking each claim to specific results from the study.

Conclusion

• The conclusion restates key findings but could be more impactful. Focus on actionable recommendations for reducing fragmentation and improving patient outcomes. For example: "Our findings highlight the urgent need to implement integrated care models and enhance care coordination to reduce mortality risks in kidney transplant patients. Policymakers should prioritise strategies to streamline healthcare delivery, particularly in fragmented systems like Colombia."

Reviewer #2: I appreciate your hard work.

Financial Disclosure:

In the "Financial Disclosure" section of PLOS One Submission System, part please add where the partial funders played specific roles, like- Study design, Data collection and analysis, Decision to publish or Preparing the manuscript. If the funders have no role you can simply write it as, "The funders did not contribute in study design, fieldwork, data analysis, decision to publish or preparation of the manuscript." But if they do, please mention in which specific sector they contributed.

Introduction:

As per your statement, "To our knowledge, no studies have specifically evaluated the association between healthcare fragmentation and survival outcomes in renal transplant recipients in low- and middle-income countries (LMICs)." So, while you have mentioned that we don't have data in LMICs, but is there any in HICs? please look for the available studies on similar studies conducted in High Income Countries so that a contrast can be build up in Discussion section. If there is no studies available in HICs you can mention it. That will give stronger rationale for the study.

Result:

Deeper analysis and findings are illustrated with well structured and readable figures and tables.

Discussion:

As mentioned, please find some existing studies conducted in High Income Setting so that this study can compare and contrast key features and increase its strength.

Acknowledgement:

Kindly remove the funding source in the "Acknowledgments" or anywhere else in the manuscript file. Funding information should only be entered in the "financial disclosure" section of the submission system as mentioned earlier in the review.

Reviewer #3: Introduction

• The introduction provides context on healthcare fragmentation and why it is an important issue.

• The literature review is relevant and establishes the need for the study.

Methods

• The study design is appropriate for the research question.

• Ethical considerations are mentioned (if applicable).

Results

• The results are well-structured and follow a logical order.

• Use of tables and figures helps in presenting data effectively.

Discussion

• The discussion links findings to the broader literature.

• The practical implications of the results are explored.

6. Conclusion

Strengths:

• Summarizes key findings effectively.

Reviewer #4: This study explores the link between healthcare fragmentation during the first year after kidney transplantation and three-year mortality risk in Colombia. The analysis of national administrative data revealed that higher fragmentation significantly increases mortality risk, even when adjusting for age, gender, region, comorbidities, and transplant year, with a 49% higher risk for those experiencing high fragmentation.

Building on the teams' prior research, this study highlights the importance of care coordination for chronic disease patients. Its findings are especially relevant for low- and middle-income countries (LMICs), where healthcare systems tend to be fragmented and under-resourced.

Strengths:

The manuscript is clearly structured and written with clarity.

The use of a national database enhances the study’s generalizability and policy relevance.

The findings are communicated in an accessible manner, with appropriate use of tables and figures.

Weaknesses:

The arguments could be better situated within the existing literature, especially regarding kidney transplantation and healthcare fragmentation in comparable contexts.

Overall Recommendation:

This is a well-written and policy-relevant manuscript addressing a significant public health issue. I recommend acceptance pending revision of the points below.

Major Issues

Methods

Lines 148–150- The authors should elaborate on how fragmentation scores were derived from the database. Were they based on the specific services provided? How are services that require multiple health workers accounted for? Although the authors have referenced previous studies that used this measure, further information is necessary to clarify its validity for their current research.

Lines 154-155- More detail is needed regarding Colombia’s insurance system and how the geographic regions were categorized. Also more information on the Charlson Comorbidity Index (CCI) is important for the general audience, a brief description of how it is calculated, what conditions it includes, and its relevance to transplant populations would enhance clarity, especially for readers outside the clinical field.

Results

Table 2- Each region should have its own unique subscript, such as "a," "b," and "c," with explanations in the footnotes as before. Bogotá, D.C. should be labeled as "Metropolitan" if more than one region is combined here (it seems so).

Discussion

Line 261-267- The authors compare their findings to research on other chronic conditions. However, the management and care coordination needs of kidney transplant patients differ substantially from those with cancers for example. The argument would be more persuasive if supported by literature on kidney transplantation, As the authors mentioned that their work is the first in LMICs, it is important to consider findings related to kidney transplants from non-LMICs. Comparing their results with studies specifically focused on kidney transplants could provide valuable context. The cited study may not be sufficient to align their findings with existing literature. This is a crucial point that needs to be addressed.

Line 267-270- The manuscript references general issues with fragmentation in other countries but lacks detail about Colombia’s healthcare system. Are delays in treatment, duplicative services, and poor care coordination documented problems locally?

Limitations- The authors noted key limitations in their study related to clinical information important for patient survival. However, they did not specify details that could confound or interact with their results. this information is for the study's conclusion.

References- Reference 13 appears to be already published, and the DOI provided is incorrect. The same article seems to be cited again as Reference 15. Please verify and adjust accordingly.

Minor Issues

Line 79–81: A citation is needed to support the description of Colombia’s healthcare system.

Lines 81–86: Important claims are made to justify the research question, but no supporting literature is cited.

Line 89–90: Citing similar studies from high-income countries would strengthen the rationale for the study.

Line 104–106: It is unclear whether the study truly addresses all the claims made in this paragraph.

Line 169–170: I suggest including the global test p-value from the Schoenfeld residuals.

Line 173: Consider adding a STROBE checklist as supplementary material.

Lines 179–181: These participant exclusion criteria should be moved to the Methods section. The rationale for excluding individuals <18 years should be stated. If this is due to different survival patterns, why not also consider an upper age limit (e.g., >75)?

Table 1: Clarify “Mean of different healthcare; X (SD)” in a footnote, please also, explain what is meant by “other department” under the region variable.

Table 2: Same labeling issues as noted above. Ensure clarity in regional categories.

Table 3 and 4: Statistically significant values should be marked (e.g., using asterisks or bold text) for easier interpretation.

Results – Were there significant differences in unadjusted models? If so, consider including or mentioning these results.

6. PLOS authors have the option to publish the peer review history of their article (what does this mean? ). If published, this will include your full peer review and any attached files.

**Do you want your identity to be public for this peer review?** For information about this choice, including consent withdrawal, please see our Privacy Policy .

Reviewer #1: **Yes: ** Faizul Akmal Abdul Rahim

Reviewer #2: **Yes: ** Rehnuma Abdullah

Reviewer #3: **Yes: ** Abdulmalik Alilu Abubakar

Reviewer #4: No

---

## [Author Response · Author response to Decision Letter 1]

4 Jun 2025

Response to reviewers

Oriana Rivera-Lozada de Bonilla

Academic Editor

PLOS ONE

Manuscript Title: Association of healthcare fragmentation and overall survival in patients with kidney transplant in Colombia

Manuscript ID: PONE-D-24-57142

Corresponding Author: Giancarlo Buitrago

Dear Editors and Reviewers,

We sincerely thank you for the valuable feedback provided on our manuscript. Your thoughtful comments and suggestions have enhanced the work's clarity, rigour, and overall quality. Below, we address each comment individually, providing detailed explanations and outlining the specific changes made to the manuscript.

Editor comments

We sincerely thank the Editor for the comments and suggestions provided. In response, we have revised the manuscript to ensure it adheres fully to PLOS ONE’s formatting and style requirements. Specifically, we have ensured that the ethics statement includes all necessary details regarding data use and approvals (Supplementary file 1). In addition, all funding information has been removed from the main text and appropriately included in the cover letter under the “Role of the Funder” section.

With respect to Figure 2, we would like to clarify that it was generated using coordinate data provided by the Colombian National Statistics Department (Departamento Administrativo Nacional de Estadística – DANE). No copyrighted maps or satellite images created using proprietary sources were employed in the figure. Therefore, we respectfully suggest that copyright permissions are not required. We are grateful for all the recommendations, which have helped improve the quality and compliance of the manuscript.

Reviewer #1:

Comment No. 1: Make the title more engaging and reflective of the results. For example: “Impact of Healthcare Fragmentation on Three-Year Survival of Kidney Transplant Recipients in Colombia.”

Answer No. 1: Many thanks for your comment related to the title, we agree with the need for a more informative and engaging title and have adapted it as your suggestions. However, since our study was observational and did not use formal causal inference methods, we believe that the term “impact” may imply causality, which could be misleading. Therefore, we have revised the title to: “Association of Healthcare Fragmentation with Three-Year Survival among Kidney Transplant Recipients in Colombia.”

Comment No. 2: The abstract does not mention why healthcare fragmentation is a significant issue in Colombia or LMICs. Add a sentence explaining why fragmentation is relevant in Colombia or LMICs.

Answer No. 2: Thank you for this valuable suggestion. We have revised the abstract to include a sentence that contextualizes the relevance of healthcare fragmentation in Colombia and other low- and middle-income countries (LMICs). The revised section now reads:

“Healthcare fragmentation, defined by inadequate communication and lack of integration among providers, disrupts the continuum of care, leading to adverse clinical outcomes. Latin American countries face significant challenges in delivering integrated healthcare. This fragmentation is further exacerbated in Colombia by the decentralized structure of the healthcare system, which disperses responsibilities across multiple actors, including public and private providers, insurers, and regional authorities.”

Comment No. 3: The conclusion is clear but repetitive with the introduction. Streamline the conclusion to avoid repetition. Focus on the implications and recommendations.

Answer No. 3: We appreciate this observation. To avoid redundancy and enhance the clarity of our conclusions, we have revised the final sentence of the abstract. The updated version reads:

“These findings underscore the importance of integrated care models and improved coordination among providers to enhance patient outcomes, particularly in resource-limited settings.” This streamlined conclusion highlights the study’s implications while minimizing overlap with the introduction.

Comment No. 4: While the introduction mentions healthcare fragmentation in general, it does not provide enough context about how fragmentation specifically impacts kidney transplant patients in Colombia. Expand on why Colombia is a relevant setting for studying fragmentation and how its healthcare system's structure creates challenges.

Answer No. 4: Thank you for this insightful comment. We agree that it is important to contextualize how healthcare fragmentation specifically affects kidney transplant recipients in Colombia. To address this, we have expanded the introduction to include the following explanation: “This fragmentation is further exacerbated in Colombia by the decentralized structure of the healthcare system, which disperses responsibility across multiple actors, including public and private providers, insurers, and regional authorities. For complex medical procedures such as kidney transplantation, once the surgery is completed, the insurer determines which providers will deliver follow-up care, including medical appointments, laboratory tests and follow-up services. Some insurers maintain agreements with specialized centers, allowing for care within a single institution during the first 6 to 12 months post-transplant. Others, however, lack such arrangements, resulting in patients receiving care from multiple providers. In these cases, the consequences of fragmentation are particularly severe as they encounter obstacles such as inconsistent follow-up, and insufficient communication between transplant centres and other providers. These challenges not only hinder recovery but also increase the likelihood of complications, hospital readmissions, and preventable mortality”

Comment No. 5: The phrase "fragmentation manifests both horizontally... and vertically" could be explained more clearly for a general audience.

Answer No. 5: Thank you for highlighting this opportunity to improve clarity. We have revised the sentence for broader accessibility, as follows:

“In Colombia, fragmentation manifests as a lack of coordination both across providers at the same level of care (horizontal fragmentation) and between different levels of care and administrative entities such as insurers and public institutions (vertical fragmentation).”

Comment No. 6: The definition of healthcare fragmentation ("measured by the number of different healthcare providers") is too simplistic and lacks methodological depth. Provide more detail on how fragmentation was quantified.

Answer No. 6: Thank you for this important observation. We have revised the Methods section to provide a more detailed explanation of how healthcare fragmentation was measured. The updated text now reads: “To measure healthcare fragmentation, we counted the number of unique healthcare providers involved in a patient's care during the first year following kidney transplantation. This approach, used in prior research, serves as a proxy for fragmented care by reflecting the dispersion of services across different providers. Each provider code corresponds to a distinct institution or healthcare facility as recorded in the administrative claims database. While this metric does not capture coordination quality directly, higher counts are assumed to indicate lower care integration.”

Comment No. 7: The rationale for selecting the 75th percentile as the cut-off for high fragmentation is not explained. Justify the use of the 75th percentile for defining high fragmentation.

Answer No. 7: Thank you for pointing out the need to clarify this methodological decision. We have now included the following justification in the manuscript:

“We defined high fragmentation as having 11 or more unique providers during the first year post-transplant, corresponding to the 75th percentile of the fragmentation distribution. This threshold was chosen to identify individuals exposed to a substantially higher-than-average degree of fragmentation. Similar percentile-based approaches have been used in health services research to define extreme exposure groups.”

Comment No. 8: Descriptions of geographical fragmentation are overly detailed, making it difficult to identify key findings. Focus on the most important findings and present them concisely. Focus on the most important findings and present them concisely. For example: "Patients with high fragmentation (≥11 providers) had a significantly higher mortality rate (18%) compared to those with low fragmentation (12%) (p=0.04)."

Answer No. 8: Thank you for your suggestion. To enhance clarity and conciseness, we have summary the geographical fragmentation results and we focused the narrative on key results related to mortality.

Comment No. 9: Some results, such as "high fragmentation was associated with a 49% increased mortality risk," are presented without interpretation or context. Briefly explain the implications of key results, e.g.: "This suggests that greater provider involvement, without effective care coordination, negatively impacts patient outcomes."

Answer No. 9: Thank you for your observation. We agree on the importance of contextualizing our findings. To address this, we expanded the discussion to include the following interpretation:

“This study evaluated the association between healthcare fragmentation during the first year after kidney transplantation and survival outcomes. Our findings demonstrate that high fragmentation, defined as exposure to 11 or more unique healthcare providers within the first-year post-transplant, was associated with a 49% increased mortality risk within three years (HR: 1.49; 95% CI 1.12–1.97; p=0.01) for each additional provider participating in healthcare during the first-year post-transplant. This suggests that a greater number of providers, in the absence of effective care coordination, may compromise continuity of care and adversely impact patient outcomes.”

Comment No. 10: Some terms (e.g., "fragmentación") appear in Spanish (Table 1), which is inconsistent with the rest of the English manuscript. Standardize the language to ensure all terms are in English.

Answer No. 10: Thank you for pointing this out. We have carefully reviewed and corrected all language inconsistencies in the manuscript. Specifically, in Table 1, the term “fragmentación” has been replaced with “fragmentation,” and we ensured that all terminology is now presented consistently in English throughout the text.

Comment No. 11: Some statements, such as "fragmentation leads to poor outcomes," are too broad and not linked directly to the study findings. Link findings more explicitly to practical implications.

Answer No. 11: Thank you for this observation. We revised the relevant section in the discussion to enhance specificity and align it more closely with our findings. We now state:

“Our findings are consistent with the existing literature, which also reports lower survival rates and higher healthcare costs among patients exposed to higher levels of fragmentation.”

Comment No. 12: The discussion does not sufficiently address how the findings can be used to inform healthcare policies or improve care coordination in Colombia. Discuss how the findings align or differ from those in other countries to emphasize the study's contribution to global literature.

Answer No. 12: Thank you for pointing out this important aspect. In response, we have added a section discussing the implications for healthcare policy and how our findings fit within the international evidence base. The revised text reads: “These findings have direct implications for health policy in countries with fragmented healthcare delivery systems, such as Colombia. Our results underscore the urgent need to implement coordinated care models and standardized referral pathways for kidney transplant recipients. In Colombia, since 2015, national efforts have been underway to standardize the management of priority health conditions through the implementation of Comprehensive Healthcare Service Pathways (Rutas Integrales de Atención en Salud, RIAS). These pathways aim to improve care coordination and ensure continuity across different levels of care. Our study contributes novel evidence from a middle-income country and highlights the need to adapt and extend integration strategies such as RIAS to complex, high-risk populations like kidney transplant recipients.”

Comment No. 13: The study could benefit from comparing findings with fragmentation studies in high-income countries to provide a broader context. Avoid overly general statements by linking each claim to specific results from the study.

Answer No. 13: To our knowledge, this is the first study to examine the association between healthcare fragmentation and three-year survival among kidney transplant recipients in both low- and high-income country settings. Previous studies in transplant populations have primarily focused on the impact of appointment adherence on graft function and patient survival. However, to date, no research has evaluated care from a healthcare system perspective, specifically addressing fragmentation as done in our study. While healthcare fragmentation has been explored in other chronic conditions, such as liver transplantation and cancer, our findings are consistent with prior literature showing that higher fragmentation is associated with lower survival rates and increased healthcare costs. Mechanisms driving these associations likely include delays in treatment, duplication of medical services, and poor coordination among healthcare providers, all of which undermine the potential benefits of transplantation.”

Comment No. 14: The conclusion restates key findings but could be more impactful. Focus on actionable recommendations for reducing fragmentation and improving patient outcomes. For example: "Our findings highlight the urgent need to implement integrated care models and enhance care coordination to reduce mortality risks in kidney transplant patients. Policymakers should prioritise strategies to streamline healthcare delivery, particularly in fragmented systems like Colombia."

Answer No. 14: Thank you for the suggestion. We have revised the conclusion to emphasize concrete recommendations and future research priorities. The updated conclusion now reads: “Our findings emphasise the importance of integrated healthcare models and enhance care coordination to reduce fragmentation, mortality risks, and optimise resource use in kidney transplant patients. Policymakers should prioritise strategies to streamline healthcare delivery, particularly in resource-limited settings and fragmented systems like Colombia. Further research is needed to explore effective strategies for integrating care and addressing regional disparities in healthcare access, which could inform policies to improve outcomes for patients with chronic conditions, including those requiring renal transplantation.”

Reviewer #2:

I appreciate your hard work.

Comment No. 1: In the "Financial Disclosure" section of PLOS One Submission System, part please add where the partial funders played specific roles, like- Study design, Data collection and analysis, Decision to publish or Preparing the manuscript. If the funders have no role you can simply write it as, "The funders did not contribute in study design, fieldwork, data analysis, decision to publish or preparation of the manuscript." But if they do, please mention in which specific sector they contributed.

Answer No. 1: Thank you for your guidance. We have revised the financial disclosure section in the submission system to specify the role of the funders. As the funders had no involvement in the study design, data collection and analysis, decision to publish, or manuscript preparation, the updated statement now reads:

“The funders did not contribute to the study design, fieldwork, data analysis, decision to publish, or preparation of the manuscript.”

Comment No. 2: As per your statement, "To our knowledge, no studies have specifically evaluated the association between healthcare fragmentation and survival outcomes in renal transplant recipients in low- and middle-income countries (LMICs)." So, while you have mentioned that we don't have data in LMICs, but is there any in HICs? please look for the available studies on similar studies conducted in

---

## [Decision Letter · Decision Letter 1]

16 Jul 2025

Association of Healthcare Fragmentation with Three-Year Survival among Kidney Transplant Recipients in Colombia

PONE-D-24-57142R1

Dear Dr. Giancarlo Buitrago,

We’re pleased to inform you that your manuscript has been judged scientifically suitable for publication and will be formally accepted for publication once it meets all outstanding technical requirements.

Kind regards,

Oriana Rivera-Lozada de Bonilla

Academic Editor

PLOS ONE

Reviewers' comments:

Reviewer's Responses to Questions

**Comments to the Author**

1. If the authors have adequately addressed your comments raised in a previous round of review and you feel that this manuscript is now acceptable for publication, you may indicate that here to bypass the “Comments to the Author” section, enter your conflict of interest statement in the “Confidential to Editor” section, and submit your "Accept" recommendation.

Reviewer #5: All comments have been addressed

2. Is the manuscript technically sound, and do the data support the conclusions?

Reviewer #5: Yes

3. Has the statistical analysis been performed appropriately and rigorously? 

Reviewer #5: Yes

4. Have the authors made all data underlying the findings in their manuscript fully available?

Reviewer #5: Yes

5. Is the manuscript presented in an intelligible fashion and written in standard English?

Reviewer #5: Yes

6. Review Comments to the Author

Reviewer #5: The revised manuscript is substantially improved and demonstrates a high level of scientific rigor, clarity, and relevance. The authors have thoroughly and thoughtfully addressed all comments from the previous review round. Specifically:

The title and abstract have been appropriately refined to avoid causal language while improving clarity and relevance.

The methodology is now clearly described, with a well-justified fragmentation metric, robust covariate adjustment, and verification of proportional hazards assumptions using Schoenfeld residuals.

The results are well-structured and statistically rigorous, with both unadjusted and adjusted findings presented transparently.

The discussion has been strengthened by including comparisons with high-income country literature and better linking findings to practical healthcare and policy implications.

Language use throughout the manuscript is clear, consistent, and adheres to standard academic English.

Data availability, ethical approval, and formatting are all fully compliant with journal requirements.

Overall, the study offers valuable insights into the impact of healthcare fragmentation on kidney transplant survival in LMIC settings and represents a meaningful contribution to the literature on health system performance and transplant outcomes.

I have no further concerns and recommend acceptance.

7. PLOS authors have the option to publish the peer review history of their article (what does this mean? ). If published, this will include your full peer review and any attached files.

**Do you want your identity to be public for this peer review?** For information about this choice, including consent withdrawal, please see our Privacy Policy .

Reviewer #5: **Yes: ** Ibrahim A. Abdulganiyyu PhD

---

## [Editor Report · Acceptance letter]

PONE-D-24-57142R1

PLOS ONE

Dear Dr. Buitrago,

I'm pleased to inform you that your manuscript has been deemed suitable for publication in PLOS ONE. Congratulations! Your manuscript is now being handed over to our production team.

Kind regards,

on behalf of

Dr. Oriana Rivera-Lozada de Bonilla

Academic Editor

PLOS ONE